# Synergistic Activity of Remdesivir–Nirmatrelvir Combination on a SARS-CoV-2 In Vitro Model and a Case Report

**DOI:** 10.3390/v15071577

**Published:** 2023-07-19

**Authors:** Anna Gidari, Samuele Sabbatini, Elisabetta Schiaroli, Sabrina Bastianelli, Sara Pierucci, Chiara Busti, Lavinia Maria Saraca, Luca Capogrossi, Maria Bruna Pasticci, Daniela Francisci

**Affiliations:** 1Department of Medicine and Surgery, Clinic of Infectious Diseases, “Santa Maria della Misericordia” Hospital, University of Perugia, 06123 Perugia, Italy; elisabetta.schiaroli@unipg.it (E.S.); sabrina.bastianelli@unipg.it (S.B.); sara.pierucci@unipg.it (S.P.); chiarabusti93@gmail.com (C.B.); lucacapogrossi@gmail.com (L.C.); mariabruna.pasticci@unipg.it (M.B.P.); daniela.francisci@unipg.it (D.F.); 2Clinic of Infectious Diseases, “Santa Maria” Hospital, Terni, 05100 Terni, Italy; lmsaraca@aospterni.it; 3Department of Medicine and Surgery, Medical Microbiology Section, University of Perugia, 06123 Perugia, Italy; samuele.sabbatini@unipg.it

**Keywords:** COVID-19, SARS-CoV-2, remdesivir, nirmatrelvir, variant, synergy, antiviral, immunosuppressed

## Abstract

Background: This study aims to investigate the activity of the remdesivir–nirmatrelvir combination against Severe Acute Respiratory Syndrome Coronavirus-2 (SARS-CoV-2) and to report a case of Coronavirus Disease 2019 (COVID-19) cured with this combination. Methods: A Vero E6 cell-based infection assay was used to investigate the in vitro activity of the remdesivir–nirmatrelvir combination. The SARS-CoV-2 strains tested were 20A.EU1, BA.1 and BA.5. After incubation, a viability assay was performed. The supernatants were collected and used for viral titration. The Highest Single Agent (HSA) reference model was calculated. An HSA score >10 is considered synergic. Results: Remdesivir and nirmatrelvir showed synergistic activity at 48 and 72 h, with an HSA score of 52.8 and 28.6, respectively (*p* < 0.0001). These data were confirmed by performing supernatant titration and against the omicron variants: the combination reduced the viral titer better than the more active compound alone. An immunocompromised patient with prolonged and critical COVID-19 was successfully treated with remdesivir, nirmatrelvir/ritonavir, tixagevimab/cilgavimab and dexamethasone, with an excellent clinical–radiological response. However, she required further off-label prolonged therapy with nirmatrelvir/ritonavir until she tested negative. Conclusions: Remdesivir–nirmatrelvir combination has synergic activity in vitro. This combination may have a role in immunosuppressed patients with severe COVID-19 and prolonged viral shedding.

## 1. Introduction

Following advice during the 15th meeting of the International Health Regulations (IHR) Emergency Committee regarding the coronavirus 2019 disease (COVID-19) pandemic, the World Health Organization (WHO) Director-General has recently determined that COVID-19 is now an established and ongoing health issue, which no longer constitutes a public health emergency of international concern [1]. However, Severe Acute Respiratory Syndrome Coronavirus 2 (SARS-CoV-2) is still circulating among the population, causing hospitalization and deaths, mainly of frail patients. Currently, the prevalent SARS-CoV-2 circulating strains belong to the Omicron variant sub-lineage, called XBB.1.5. This variant is now considered a variant of interest (VOI), with no evidence of higher severity of associated disease but increased transmissibility [2]. 

A screenshot of the situation in Italy shows more than 60,000 hospitalized patients and 458 deaths in the last 30 days [3], highlighting that COVID-19 still engages health care in an incisive way.

Of particular relevance are elderly and immunocompromised patients, of which those with hematologic malignancy show a high mortality rate. The greater susceptibility is caused by the lack of response to vaccination and the consequent insufficient antibody response against SARS-CoV-2 [4]. Even if the correlation of humoral immunity with protection against SARS-CoV-2 is still debated [5], the lack of a fully working immune system in hematologic patients leads to increased risk for severe COVID-19, prolonged positivity and persistent disease [6,7].

The current therapies available for COVID-19 mainly involve the use of antivirals and, in some cases, monoclonal antibodies [8,9]. These include nucleoside analogues, such as remdesivir and molnupiravir, and a viral protease inhibitor nirmatrelvir given in combination with ritonavir [10]. However, since March 2023, molnupiravir has no longer been available for prescription, as the “Agenzia Italiana del Farmaco” (AIFA) stated [11]. This decision followed the lack of demonstration of mortality and hospital admission reduction in COVID-19 patients treated with molnupiravir [11]. Similar drugs were initially used against HIV, and thanks to the already acquired knowledge in this field, it was possible to develop new ones in a very short time [12]. Precisely, in the fight against HIV, combined therapies have been shown to be much more effective in containing the infection and guaranteeing total inhibition of viral replication [13].

The aim of this study was to test, for the first time, the combination of remdesivir and nirmatrelvir on a cell-based SARS-CoV-2 infection model to find a possible synergic effect against viral replication. Furthermore, we described a case report of the real-life use of this combination on a hematologic patient being treated at “Santa Maria” hospital in Terni, Italy.

## 2. Materials and Methods

### 2.1. SARS-CoV-2 Strains, Vero E6 Cell Cultures and Compounds

All the SARS-CoV-2 strains used in the experiments were isolated in Biosafety Level 3 (BSL3) Virology laboratory at “Santa Maria della Misericordia Hospital”, Perugia, Italy, as previously described [14]. Vero E6 cells were cultured in Eagle’s minimum essential medium (EMEM), supplemented with 10% foetal bovine serum (FBS) and 1% penicillin-streptomycin at 37 °C with 5% CO_2_ and used for viral replication. The culture supernatants were titered via Half-maximal Tissue Culture Infectious Dose (TCID50) endpoint dilution assay [15], aliquoted and stored at −80 °C.

The full-length SARS-CoV-2 viral genome was obtained using the COVIDSeq Assay (Illumina MiSeq Instrument, San Diego, CA, USA) at the Virology laboratory of the Department of Medical Biotechnologies, University of Siena, Siena, Italy, as previously described [16]. Whole-genome sequencing was submitted to GISAID (http://gisaid.org/) for variant assignment. The strain used for the experiments was a SARS-CoV-2 20A.EU1 (lineage B.1) clustered with viruses circulating in Europe from spring to the end of 2020, Omicron sub lineage BA.1 isolated on 5 September 2022, from the patient in the case report and sub-lineage BA.5 isolated from a symptomatic patient on 4 June 2022. Viral stock aliquots were thawed immediately before each experiment and discarded after use.

Remdesivir (Veklury^®^, Gilead, Foster City, CA, USA) and nirmatrelvir (PF-07321332, MedChemExpress, Monmouth Junction, NJ, USA) were reconstituted using sterile water and dimethyl sulfoxide (DMSO), respectively, and stored at −80 °C. Before each experiment, frozen aliquots were diluted with complete medium to reach the desired concentration.

### 2.2. Determination of Remdesivir Effective Concentrations 50 and 90 (EC50 and EC90)

The effect of remdesivir against SARS-CoV-2 strains was assessed through yield reduction assay, as previously described with some modifications [17,18]. Briefly, Vero E6 cells were seeded in 96-well clear flat-bottom plates and incubated at 37 °C with 5% CO_2_ for 24 h. After incubation, cells were mock infected or infected using SARS-CoV-2 (multiplicity of infection 0.1) and subsequently treated with different concentrations of remdesivir (0.62–50 µM). Remdesivir-treated cells (Negative controls) were included in each plate. Cell viability was measured using the 3-[4,5-dimethylthiazol-2-yl]-2,5-diphenyl tetrazolium bromide (MTT) reduction assay [17] after 48 and 72 h of incubation. Briefly, 100 µL of MTT solution (0.5 mg/mL) was added to the cells for 3 h in the same conditions and formazan crystals were dissolved with 100 µL of DMSO for 1 h. The absorbance was read with a microplate reader (Tecan Infinite M200, Tecan Trading AG, Männedorf, Switzerland) at 570 nm setting the reference filter at 630 nm.

EC50 and EC90 were obtained by fitting drug dose–response curves by means of variable slope regression modelling.

The experiment was repeated thrice with three technical replicates.

### 2.3. Drug Combination Test

Nirmatrelvir EC50 and EC90 values were obtained from our previous paper [18]. The synergistic effect of remdesivir and nirmatrelvir combinations was determined as previously described [19]. The combinations were tested through yield reduction assay, treating the infected cells with serial dilutions of each antiviral starting from the EC90 previously obtained. Cells were incubated with 0.1 MOI of virus for 1 h; then, supernatants were removed and compounds added. At 48 and 72 h of treatment, the supernatants were removed and used for viral titer determination via plaque assay. Thereafter, MTT reduction assay was used for cell viability determination. Viability recovery was calculated according to the following formula:Viability recovery % = [(treated infected cell viability − infected cell viability)]/[(mock infected cell viability − infected cell viability)] × 100

All tests were conducted in triplicate in three independent experiments.

To test whether the drug combinations act synergistically, the observed responses were compared with expected combination responses. The expected responses were calculated based on the Highest Single Agent (HSA) reference model using SynergyFinder version 2 [19]. This model starts from the consideration that a combination with more effect than the highest single agent (E_A,B,…,N_ > max (E_A_, E_B,_ E_N_)) must interact someway. Mathematically, it is based on the calculation of the HSA synergy score (S_HSA_) by using the following equation: S_HSA_ = [E_A,B,…,N_ − max(E_A_, E_B_, …, E_N_)]. An HSA score > 10 is considered synergic. When the synergy score is less than −10, the interaction between two drugs is likely to be antagonistic, while when it is from −10 to 10, the interaction is likely to be additive [19].

### 2.4. Plaque Reduction Assay

To better understand the effect on viral replication and viral particle release of treated cells after infection, SARS-CoV-2 titers were determined on selected supernatants [20]. Briefly, monolayers of Vero E6 cells were inoculated with 250 µL of ten-fold serial dilution of the supernatants obtained in drug combination tests. Once the inoculum was removed, cells were overlaid with complete medium with agar 0.1% for 72 h. After incubation, monolayers were fixed and stained with a solution containing 4% formalin and 0.5% crystal violet. Wells showing 2 to 50 plaques were considered for the determination of plaque-forming units per mL (PFU/mL). Each sample was analyzed in triplicate.

### 2.5. Statistical Analysis and Data Elaboration

GraphPad 8.3 software (GraphPad Software, San Diego, CA, USA) was used for all statistical analysis. The assessment of normality was performed using the Kolmogorov–Smirnov test, and data were shown as mean with the respective standard deviation (SD) or median with interquartile range (IQR), as appropriate. All the experiments were performed at least twice with three technical replicates.

### 2.6. Case Report and In Vitro Infectivity

Here, we describe a case of a patient admitted for COVID-19 to the Clinic of Infectious Diseases of the “Santa Maria” hospital of Terni, Terni (TR), Italy. The patient provided informed oral consent for clinical data collection. The data were collected from electronic medical records (jpalio Hospital Information System, jHIS™). 

Respiratory samples (nasopharyngeal swabs) were tested for SARS-CoV-2 RNA using the Xpert^®^ Xpress SARS-CoV-2 (Cepheid, Sunnyvale, CA, USA), as previously described [21]. 

The virus isolation and titration were conducted in the biosafety level-3 Virology laboratory at “Santa Maria della Misericordia Hospital”, Perugia, Italy, as previously described [14]. 

## 3. Results

The antiviral effect and EC90 of nirmatrelvir were acquired from our previous article and used for subsequent experiments. Nirmatrelvir EC50 and EC90 after 48 h treatment were 1.28 µM and 3.70 µM, respectively. After 72 h, nirmatrelvir EC50 and EC90 were 1.75 µM and 4.46 µM, respectively [18]. Remdesivir antiviral activity was tested in vitro with a Vero E6 cell-based viability model. Four-parameter variable slope regression modelling of remdesivir dose–response showed an EC50 of 1.2 µM (95% confidence interval, CI, 0.6–2.4) and an EC90 of 5.7 µM with a slope of 1.4 (95% CI 0.6–4.8) after 48 h of treatment (Appendix A). When the incubation was extended to 72 h, remdesivir EC50 and EC90 were 1.2 µM (95% CI 0.8–1.9) and 3.1 µM, respectively, with a slope of 2.4 (95% CI 1.3 to 3.8) (Appendix A). 

The subsequent experiments with combination treatment were performed starting from the EC90s and proceeding with 2-fold serial dilutions. Vero E6 cells were previously infected with SARS-CoV-2 20A.EU1 strain and then treated with the antiviral combination. After 48 h and 72 h of incubation, viability was determined as above. 

Remdesivir and nirmatrelvir showed a synergistic activity, both at 48 h and 72 h, with an HSA score of 52.8 (±9.82) and 28.6 (±11.68), respectively (*p* < 0.0001, Figure 1A,B, Appendix A). 

Before the MTT assay, the supernatants were collected and stored for subsequent determination of the viral titer. Three concentrations from the 72 h experiment were selected to perform supernatant titration with the plaque assay.

The combination of remdesivir and nirmatrelvir reduced the viral titer significantly more than remdesivir alone (the more active compound) at the two higher concentrations (Figure 2A, *p* = 0.0003). The compounds together caused a viral titer reduction by an extra 1.6–1.8 log compared to remdesivir alone, which, in turn, reduced the viral titer by 0.3–3.3 compared to the untreated control. Nirmatrelvir reduced the viral titer of 1.3 log compared to the untreated control only at the higher concentration tested. Furthermore, the combination reduced the viral titer of 4.8–4.9 log compared to the control and, at the higher concentration, reached viral eradication (<50 PFU/mL, the detection limit of the method).

The combination was also tested on BA.1 (isolated on 5 September 2022 from the patient of the case report below) and BA.5 SARS-CoV-2 variants. As shown in Figure 2B, remdesivir at the higher concentration reduced the viral titer below the detection limit, so it was not possible to establish if the combination is more effective. The remdesivir–nirmatrelvir combination at the respective concentrations of 1.6 and 2.2 μM reduced the viral titer by 0.7 logs more than remdesivir alone (*p* = 0.048), while the lower concentrations reduced the titer by 0.1 logs more than remdesivir (*p* = 0.11). 

The combination caused an extra viral titer reduction of 0.5 logs on the BA.5 strain compared to remdesivir alone, but the differences were not significant (*p* = 0.059, Figure 2C). Similarly, for the BA.1 strain, remdesivir at the higher concentration reduced the viral titer below the detection limit.

### Case Report

A 50-year-old woman was admitted to “Santa Maria” hospital in Terni for a suspected recurrence of lymphoproliferative disease on 5 February 2022. She had a history of left adnexectomy for dermoid cyst at 19 years old and, in 2015, left adrenalectomy for pheochromocytoma and simultaneous finding of non-Hodgkin (NH) follicular lymphoma. In 2018, she had a lymphoma recurrence. She was then treated with chemoimmunotherapy (rituximab as first-line and rituximab plus bendamustine at recurrence) and developed secondary liver damage as a consequence of autoimmune hepatitis. Therefore, she was treated with steroid therapy, leading to normal liver function values.

The patient had previously received three doses of the Pfizer COVID-19 mRNA vaccine. On 18 February, she underwent a nasopharyngeal swab that came back positive (Xpert^®^ Xpress SARS-CoV-2, Cepheid) and was transferred to the Clinic of Infectious Diseases at the same hospital. The patient had few COVID-19 symptoms (dry cough) so was treated with early antiviral therapy (remdesivir for three days and sotrovimab). She remained persistently asymptomatic until 24 March, when, following a progressive worsening of the respiratory picture, she performed chest Computed Tomography (CT) angiography, which was negative for pulmonary embolism and showed “the appearance of multiple and diffuse areas of parenchymal opacities with a ground glass appearance, with a tendency to consolidation in sloping areas”. Therefore, she was treated with steroid therapy (dexamethasone 6 mg intravenous once daily for 10 days) with the resolution of the clinical picture. All exams performed to investigate the suspected lymphoproliferative disease recurrence were inconclusive. However, during all this time, multiple nasopharyngeal swabs were collected and all tested positive for SARS-CoV-2. Furthermore, on 19 April and 4 May, two nasopharyngeal swabs (positive for SARS-CoV-2) were collected, and culture-based virus isolation in Vero E6 cells was performed to evaluate the potential infectivity. SARS-CoV-2 was isolated from these two specimens, and both the viruses were sequenced as Omicron 1 variant (BA.1). She was discharged on 24 May 2022, asymptomatic but still positive for SARS-CoV-2. During the following period, the patient remained in an isolation regimen at home and performed multiple nasopharyngeal swabs, always testing positive for SARS-CoV-2.

On 5 September, a new culture-based isolation test for SARS-CoV-2 from a swab specimen was performed and this also tested positive. This virus was sequenced as the Omicron 1 (BA.1) variant, as in the previous specimens.

The patient was readmitted to the Clinic of Infectious Diseases to treat the infection with off-label antiviral therapy on 21 September 2022. The clinical conditions immediately appeared serious with acute progressive respiratory failure.

She underwent a chest CT scan, which showed “Bilateral extensive consolidative foci, “crazy paving” and “ground glass” opacities are evident in the lung parenchyma”. In addition, bronchoalveolar lavage was performed and tested positive for SARS-CoV-2, low-burden *Haemophilus parainfluenzae* and methicillin-sensitive *Staphylococcus aureus* (MSSA) 20000 CFU. It was also negative for *Pneumocystis carinii* and Cytomegalovirus. The specimen was also tested with BIOFIRE^®^ FILMARRAY^®^ Pneumonia Plus, which tested positive for MSSA.

Due to severe respiratory insufficiency, she required high-flow nasal cannula (HFNC) oxygen support and had hemodynamic instability, which required inotropic support.

She was evaluated by the resuscitator specialist who judged the prognosis as poor and defined the patient as not eligible for intensive care.

The medical therapy was set as follows: targeted antibiotic therapy, off-label antiviral therapy with remdesivir (10-day course) plus nirmatrelvir/ritonavir (5-day course), tixagevimab/cilgavimab and dexamethasone (6 mg/die for 10 days). No adverse effects were observed, except for mild bradycardia (50–60 beats per minute) on day 10 of therapy with remdesivir, which regressed after the drug discontinuation. The patient presented an excellent clinical–radiological response with a progressive reduction in the oxygen requirement up to the suspension and improvement on control CT after 2 weeks. However, despite the fact that the patient was isolated in a single room without the risk of novel SARS-CoV-2 exposure, the nasopharyngeal swab was persistently positive for SARS-CoV-2, and she required further off-label prolonged therapy with nirmatrelvir/ritonavir to reach negativization. She started this therapy on 21st October, continuing until 11 November, and the first negative swab was obtained on 2 November 2022. No severe adverse events were reported; the patient complained of mild gastric intolerance, but she completed the treatment. The patient was discharged on 6 November, asymptomatic and without the need for oxygen therapy. She underwent a subsequent nasopharyngeal swab, which tested negative a month later (Figure 3).

## 4. Discussion

After more than three years, the WHO declared the end to COVID-19 as a global health emergency [1]. This is a huge result and important acknowledgement of the intense work of the scientific community that, during the pandemic, worked tirelessly to fight and contain the spread of the virus. Compared to the beginning of the pandemic, we have various weapons at our disposal to treat and prevent the infection. Among these, vaccines and the three available antivirals (remdesivir, nirmatrelvir/ritonavir and molnupiravir) are certainly the key tools, while monoclonal antibodies and hyperimmune plasma are currently taking a backseat, especially due to the emergence of variants of concern (VOCs) [10].

However, SARS-CoV-2 is still spreading and killing, especially frail patients. Among frail patients, the immunocompromised represent one of the most difficult challenges. It is known that innate and adaptive immunity is involved in preventing, limiting and clearing SARS-CoV-2 infections [7]. Firstly they showed poor response to the vaccine, with varying levels across different types of immunodepression [22]. In particular, antibody responses to vaccines are lowest among solid organ transplant and anti-CD20 monoclonal recipients [23]. They often showed prolonged SARS-CoV-2 infections with different levels of illness severity and consequent increased morbidity and mortality [7,24]. Furthermore, conventional therapeutic schemes are often not effective in treating COVID-19 in immunocompromised populations [25,26,27]. In these patients, prolonged viral shedding may lead to antiviral resistance, already well documented about remdesivir [28,29]. The mutations involve the nsp12 RNA-dependent RNA polymerase and cause an EC50 increase that impedes viral eradication [28,29]. Combination therapy, especially if molecules with different targets are used, can prevent the onset of resistance and counteract the resistance conferred by the emerging variants. However, the international guidelines do not provide specific therapeutic protocols for these kinds of patients because there is insufficient evidence to guide clinical recommendations on using combinations of antiviral agents or on extending the duration of therapies [10]. Indeed, few data are available in the literature to support the need for specific protocols for immunocompromised patients. 

The intuition that a combination of antivirals is more effective than a single drug is certainly not new. The efficacy of combination therapies has already been widely demonstrated for other viruses, such as HIV and HCV, especially when the drugs act on different targets. Based on this, the scientific literature has begun to lay the foundations for carrying out similar treatments in fragile patients with difficult-to-treat COVID-19. There are few in vitro and animal studies, while there are no clinical trials but only retrospective clinical reports (case reports and a retrospective observational study) [25,26,27,30]. Our research group studied the combination of molnupiravir and nirmatrelvir, demonstrating the synergistic activity of this combination in vitro [18]. The greatest advantage of this combination is the possibility of administering it as an oral therapy, without the need for hospitalization. Oral administration has the potential to maximize the clinical benefits, including decreased duration of COVID-19 and reduced post-acute sequelae of SARS-CoV-2 infection, as well as limited side effects, such as hepatic accumulation [31]. Other studies have confirmed that this is a very promising combination on Omicron variants as well, both in vitro and in animal models [32,33]. To the best of our knowledge, there is only one case report describing the outcome of a patient with persistent COVID-19 treated with this combination [34]. The authors observed clinical, radiological and microbiological success in an immunosuppressed patient [34]. However, in Italy, molnupiravir is no longer available for prescription, so a different kind of drug needs to be tested in combination with nirmatrelvir [11].

A wide number of drug combinations have been tested on SARS-CoV-2 in vitro. Remdesivir seems to be synergic with anti-HCV direct-acting antivirals (DAASs) in Vero E6 in vitro models, such as ivermectin, emetine, gemcitabine, nelfinavir and amodaquine [12,32,35,36]. SARS-CoV-2 polymerase is a good antiviral target, and about 43 molecules showed synergistic activity with molnupiravir and remdesivir, especially if they have different targets. Indeed, molecules that act on the same target showed only additive effects [12]. Other combinations with potential antiviral synergistic activity are nelfinavir with molnupiravir, nelfinavir with remdesivir in Calu-3 in vitro models or molnupiravir with favipiravir in hamster and Vero E6 models [37,38].

Few data are available in in vitro and in vivo models about the combination of nirmatrelvir plus remdesivir but, paradoxically, compared to other combinations, it is the one with which we have the most clinical experience.

To the best of our knowledge, there are only two studies that explored the activity of nirmatrelvir and remdesivir in combination, but one tested these compounds only on the feline coronavirus (FCoV) biotype and not on SARS-CoV-2. The authors found synergistic activity of this combination [39]. The other study tested some concentrations of remdesivir and nirmatrelvir on SARS-CoV-2 in Vero E6 or A549-hACE2 cells. Infected cells were visualized via spike protein immunostaining. The authors observed enhanced efficacy of the combination with residual infectivity decreased to 5.7- and 3.9-fold compared to treatments with nirmatrelvir and remdesivir, respectively [40]. However, based on these results, it is not possible to establish if the interaction of the compounds is synergic or additive.

The only in vivo study available in the literature is inconclusive because remdesivir was degraded in mice plasma by the enzyme carboxylesterase 1c (Ces1c) before it reached tissues [33].

To the best of our knowledge, this is the first in vitro study that tested the synergistic activity of nirmatrelvir and remdesivir. Both the compounds showed high activity on SARS-CoV-2, with micromolar effective concentrations. According to our data, the compounds showed marked synergistic activity, independent of the time of incubation. Furthermore, at the concentration tested, the improvement in the effect is also maintained on Omicron variants. Indeed, we tested BA.5 and BA.1 variants because the latter was isolated from the patient whose case report we described. The combination was able to reduce the viral titer by an extra 1.6–1.8 logs more than the most effective single compound.

We hypothesize that the synergistic mechanism of this antiviral combination involves the inactivation of the exonuclease nsp14 that, if active, could contrast the activity of remdesivir. Nirmatrelvir inhibits the protease 3CL that is able to cleave 11 non-structural proteins including nsp14 inactivating them [41]. Consequently, without the activity of the SARS-CoV-2 exonuclease, remdesivir is not inhibited. This hypothesis needs to be demonstrated.

Clinical data supporting the validity of this combination are in the early stages. There are only retrospective observational studies available. The largest one is an Italian study published by Mikulska et al. The authors enrolled 22 patients with relapsed/prolonged COVID-19 treated with nirmatrelvir/ritonavir, remdesivir and, if available, anti-spike monoclonal antibodies (mAbs) [30]. It is noteworthy that 68% of the patients had previously received anti-CD20 therapy, a condition they share with our patient and other cases described in the literature and that predisposes them to an impaired immunological response [22,23,25,26,27,30]. Mikulska and colleagues observed that on the last follow-up day, 18/22 patients were negative and asymptomatic for SARS-CoV-2, 3 patients died, and 1 had improved symptoms but persistent antigen and polymerase chain reaction (PCR) positivity. They also observed that the rate of virological response was higher in patients treated with a combination including mAbs and in patients with previous anti-CD20 treatment. Also, the number of vaccine doses influenced the rate of response [30]. These data are in line with the other case reports described in the literature [25,26,27] and also with the case reported here. Our patient was treated with remdesivir, nirmatrelvir/ritonavir and tixagevimab/cilgavimab, associated with antibiotic therapy and dexamethasone. This therapy led to impressive radiological and clinical improvement, but it was not enough to eradicate the infection after more than 7 months of positivity. Most likely, a longer course of therapy could have been more effective to reach viral eradication. Indeed, she needed a prolonged course of nirmatrelvir/ritonavir to reach a stable negativity. 

This study has a few limitations, such as the in vitro model, which could overestimate or underestimate the antiviral efficacy, the lack of other less common variants and the use of non-human cell lines. The use of an in vitro model does not consider the role of the immune system and the pharmacokinetics of the drugs, such as the use of a non-human cell line. However, for the tested compounds, these aspects are known in the literature. Furthermore, the synergistic checkerboard on omicron variants was not performed because we measured the antiviral effect as cell viability recovery, and it was not possible to perform this test on the omicron variants for their less efficient propagation on cell lines, such as Vero E6 and Calu-3 [32]. Therefore, some combination concentrations included in the most synergistic area were tested on Omicron 1 and 5 measuring the viral titer reduction. Likewise, a single case report is not sufficient to validate a protocol therapy, but every experience could help physicians to manage difficult-to-treat COVID-19. The patient was also treated with other therapies apart from the combination nirmatrelvir/ritonavir and remdesivir. It makes it difficult to establish the real role of the combination, but it is a problem often met in real-life experiences [25,26].

Further studies are necessary to confirm the effectiveness of this combination, especially in vivo and hopefully in a clinical trial.

## 5. Conclusions

The remdesivir–nirmatrelvir combination showed good synergic activity in vitro. This combination may have a global impact on difficult-to-treat and severe COVID-19, especially in immunocompromised patients that often present prolonged relapsed COVID-19. Further studies are absolutely necessary to confirm these preliminary data.

## Figures and Tables

**Figure 1 viruses-15-01577-f001:**
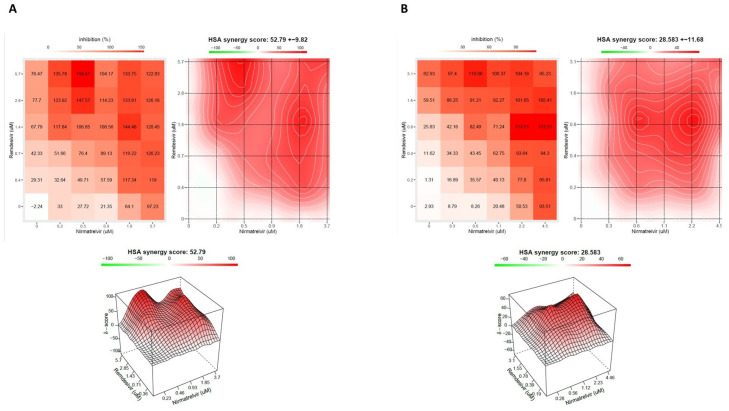
The interaction landscapes of the remdesivir–nirmatrelvir combination after 48 h (**A**) and 72 h (**B**) of incubation. A Vero E6 cell-based infection assay was used to investigate the in vitro activity of the combination. All experiments were performed using a 20A.EU1 Severe Acute Respiratory Syndrome Coronavirus 2 (SARS-CoV-2) strain. After the incubation, a viability assay was performed. To test whether the drug combinations act synergistically, the Highest Single Agent (HSA) reference model was calculated. An HSA score > 10 is considered synergic. Data are from 3 independent experiments performed in triplicate. Both panels (**A**,**B**) show, in order, numerical values across the matrices, synergy map and 3D plot.

**Figure 2 viruses-15-01577-f002:**
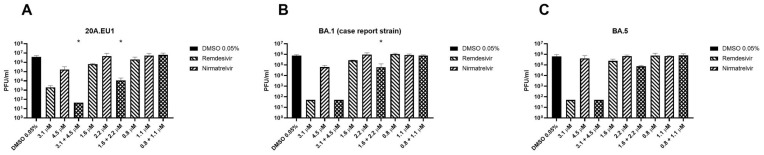
Viral titration (plaque reduction assay) of remdesivir–nirmatrelvir combination. After synergy tests (**A**) or yield reduction assay (**B**,**C**), supernatants of SARS-CoV-2 20A.EU1 (**A**), BA.1 (case report strain) (**B**) and BA.5 (**C**) infected Vero E6 cells treated with the antiviral combination were frozen and tested for viral load. Data are expressed as plaque-forming units (PFU)/mL (mean ± standard deviation, SD) from 2 or 3 experiments performed in triplicate. * *p* < 0.05, antiviral combination vs more active single agent. Viral eradication was defined as <50 PFU/mL, the detection limit of the method. (**A**): The combination at the concentration of 3.1 + 4.5 μM eradicated the virus. (**B**,**C**): remdesivir at the concentration of 3.1 μM and the combination at the concentration of 3.1 + 4.5 μM eradicated the virus.

**Figure 3 viruses-15-01577-f003:**
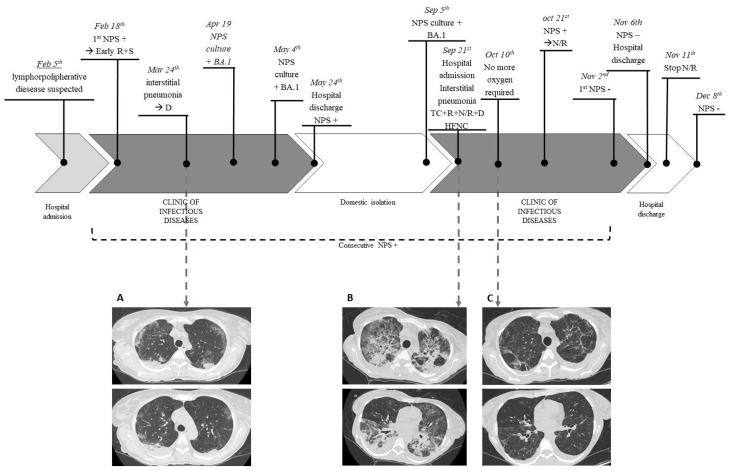
Case report timeline: A 50-year-old woman with a history of non-Hodgkin follicular lymphoma treated with anti-CD20 therapy was SARS-CoV-2 positive starting on 18 February 2022. Multiple nasopharyngeal swabs were collected and all tested positive. On 19 April, 4 May and 5 September, SARS-CoV-2 was isolated from nasopharyngeal swabs, and the viruses were identified as BA.1 variants. On 24 March, she underwent Computed Tomography (CT) that showed “multiple and diffuse areas of parenchymal opacities with a ground glass appearance, with a tendency to consolidation in sloping areas” (**A**). On hospital readmission, the clinical conditions immediately appeared critical with acute progressive respiratory failure due to extensive COVID-19 interstitial pneumonia, as shown in CT images (**B**): “Bilateral extensive consolidative foci, “crazy paving” and “ground glass” opacities are evident in the lung parenchyma””. The medical therapy was set as follows: antibiotic therapy, off-label antiviral therapy with remdesivir (10-day course) plus nirmatrelvir/ritonavir (5 day course), tixagevimab/cilgavimab, and dexamethasone (6 mg/die for 10 days). The patient presented an excellent clinical-radiological response with progressive reduction of oxygen requirement up to the suspension and improvement on control CT after 2 weeks (**C**). However, she required further off-label prolonged therapy with nirmatrelvir/ritonavir up to negativization obtained on 2 November 2022. NPS, nasopharyngeal swab; R, remdesivir; S, sotrovimab; D, dexamethasone; TC, tixagevimab/cilgavimab; N/R, nirmatrelvir/ritonavir; HFNC, high-flow nasocanula.

## Data Availability

The datasets used and analyzed during the current study are available from the corresponding author on reasonable request.

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
