# Peer review of "Synergistic Activity of Remdesivir–Nirmatrelvir Combination on a SARS-CoV-2 In Vitro Model and a Case Report"

_viruses, 2023, doi:10.3390/v15071577_

Round 1
Reviewer 1 Report
The article "Remdesivir-nirmatrelvir combination: in vitro activity and a case report." attempts to highlighting the potential value of combination therapy of nirmatrelvir/remdesivir in improving antiviral activity. The paper is well structured. I recommend this article to be published in the journal. Here are some suggestions:
1. Take page 7 as an example. There are too many paragraphs in the article. Please adjust them.
2. Please add relevant literatures. For example, in line 62, “The current therapies available for COVID-19 mainly involve the use of antivirals. (DOI: 10.3390/ijms24108867; DOI: 10.1016/j.ejmech.2023.115503)”; in lines 69-70, “it has been seen that combined therapies (DOI: 10.3390/biomedicines9060689) are much more effective in containing the infection and guaranteeing total inhibition of viral replication.”
3. In “Discussion”, As new variants continue to evolve, it is important to discuss the drug resistance of Remdesivir. In fact, Remdesivir drug resistance has been extensively studies. Please refer to: DOI: 10.1038/s41467-022-29104-y. Combination therapy could address drug resistance conferred by emerging variants.
4. Check the abbreviations throughout the manuscript and introduce the abbreviation when the full word appears the first time in the abstract and the remaining for the text and then use only the abbreviation (For example, WHO). Make a word abbreviated in the article that is repeated at least two times in the text, not all words to be abbreviated (For example, IHR, VOCs).
5. Oral administration could be considered as game-changers in treating COVID-19. In line 331, for the benefits of the readers, please supply relevant knowledge: “Oral administration has the potential to maximize clinical benefits, including decreased duration of COVID-19 and reduced post-acute sequelae of SARS-CoV-2 infection, as well as limited side effects such as hepatic accumulation. (DOI: 10.3389/fimmu.2022.1015355)”
Minor editing of English language required
Author Response
The article "Remdesivir-nirmatrelvir combination: in vitro activity and a case report." attempts to highlighting the potential value of combination therapy of nirmatrelvir/remdesivir in improving antiviral activity. The paper is well structured. I recommend this article to be published in the journal. Here are some suggestions:
- Take page 7 as an example. There are too many paragraphs in the article. Please adjust them.
Thank you for the suggestion. As you recommended, the text has been grouped into fewer paragraphs.
- Please add relevant literatures. For example, in line 62, “The current therapies available for COVID-19 mainly involve the use of antivirals. (DOI: 10.3390/ijms24108867; DOI: 10.1016/j.ejmech.2023.115503)”; in lines 69-70, “it has been seen that combined therapies (DOI: 10.3390/biomedicines9060689) are much more effective in containing the infection and guaranteeing total inhibition of viral replication.”
Thank you for the suggestion, we added the references you requested.
- In “Discussion”, As new variants continue to evolve, it is important to discuss the drug resistance of Remdesivir. In fact, Remdesivir drug resistance has been extensively studies. Please refer to: DOI: 10.1038/s41467-022-29104-y. Combination therapy could address drug resistance conferred by emerging variants.
Thank you for the comment. We added the discussion of the drug resistance onset of remdesivir (lines 369-374).
- Check the abbreviations throughout the manuscript and introduce the abbreviation when the full word appears the first time in the abstract and the remaining for the text and then use only the abbreviation (For example, WHO). Make a word abbreviated in the article that is repeated at least two times in the text, not all words to be abbreviated (For example, IHR, VOCs).
As you suggested, all abbreviations have been checked and corrected.
- Oral administration could be considered as game-changers in treating COVID-19. In line 331, for the benefits of the readers, please supply relevant knowledge: “Oral administration has the potential to maximize clinical benefits, including decreased duration of COVID-19 and reduced post-acute sequelae of SARS-CoV-2 infection, as well as limited side effects such as hepatic accumulation. (DOI: 10.3389/fimmu.2022.1015355)”
Thank you for your comment, the citation has been added (lines 390-392).
Reviewer 2 Report
The manuscript by Gidari et al., documents the details of in vitro activity of Remdesivir-nirmatrelvir combination and a case report.
Overall, the manuscript needs extensive English revision. Further, the manuscript has some major concerns. Remdesivir and Nirmatrelvir combination is studies for in vitro, pre-clinical as well as clinical settings. So I failed to understand the novelty of this study.
Fig.1 Remdesivir is very well studies for its cytotoxicity as well as anti-COVID activity. Hence, Fig.1 is redundant and not required.
A time of addition experiment with combination of both drugs will be required.
Figure 3 and 4: Y-axis Absorbance (OD 570 nm) is incorrect.
What are the images in the Figure 5? No labelling or description is provided.
Line #269 - 6mg/die for 10 days?????
Manuscript does not describe the MTT assay in the methods section.
Line #269 - 6mg/die for 10 days?????
Author Response
The manuscript by Gidari et al., documents the details of in vitro activity of Remdesivir-nirmatrelvir combination and a case report.
Overall, the manuscript needs extensive English revision.
Thank you for the comment. As you suggested, we submitted the manuscript to a mother tongue for English revision.
Further, the manuscript has some major concerns. Remdesivir and Nirmatrelvir combination is studies for in vitro, pre-clinical as well as clinical settings. So I failed to understand the novelty of this study.
Thank you for the comment. Actually, to the best of our knowledge, no study explored the synergistic activity of this combination on SARS-CoV-2 in vitro. As specified in the discussion, there are only two available in vitro studies, the first one is not on SARS-CoV-2 but on a feline coronavirus. The second one tested some concentration finding an improvement in the antiviral activity, without a study of the synergistic score. Furthermore, the only in vivo available study is inconclusive. No trial clinics have been conducted, the only available data are case reports and a retrospective observational study. All are cited in the discussion of our paper. So, if the combination may have a role in COVID-19 treatment, it is important to study the activity in all these settings, including the in vitro ones.
Fig.1 Remdesivir is very well studies for its cytotoxicity as well as anti-COVID activity. Hence, Fig.1 is redundant and not required.
Thank you for the suggestion. Fig. 1 has been moved to Supplementary Figure 1.
A time of addition experiment with combination of both drugs will be required.
Before deciding the timing of the experiments, we tested different time points, concluding that 48 and 72 h are the most representative to observe viability modifications after compounds and viral effects. We observed that after 24 h of incubation was too early to see significant effects and after 72 h (we tested 96 h), cell viability is naturally reduced, or cells are destroyed by the virus if not completely eradicated. For these reasons, we selected these specific time points.
Figure 3 and 4: Y-axis Absorbance (OD 570 nm) is incorrect.
Thank you for the correction. The error has been fixed (see new Figure 2 A-B-C).
What are the images in the Figure 5? No labelling or description is provided.
Thank you for the suggestion, the image descriptions have been added (see Figure 3).
Line #269 - 6mg/die for 10 days?????
The sentence has been modified (line3 282-283).
Manuscript does not describe the MTT assay in the methods section.
The MTT assay description has been added as requested (line 117-120).
Reviewer 3 Report
Gidari et al provide in vitro data showing the synergistic effects of remdesivir and nirmatrelvir and provide a case report where this combination might have contributed to the viral clearance in an immunosuppressed covid-19 patient. The amount of data provided is low, but appears valid. However, the methodology needs to be explained and referenced more thoroughly. The interpretation that this combination was required for negativization could be weakened. Nonetheless, since this combination is clinically applied in patients, the in vitro analysis provided here gives important background information.
Specific points
62 Current therapies also include monoclonal antibodies. Also the patient in the case report received several antibody treatments. This should be added here.
66-70 provide references
88-90 Provide GISAID IDs
106 in the provided reference [13] authors use Vero-GFP cells to monitor CPE. When working with other cells they use MTS instead of MTT. Vero cells do not express TMPRSS2 so the CPE is low, making determination via viability assays not very effective. However, there is differences between labs and CPE is very prominent in the author’s hands. To confirm this, microscopy images of CPE should be provided.
115 Was the inoculum also removed (wash control)? Or do the 48 h samples still contain the initially added virus?
Fig 1 Please also normalize this Figure and show as %infection or as %viability recovery, so the EC50/90 can be easily estimated from the graph. Also include the values obtained in absence of virus. Consider to show the x-axis as log scale. Do the determined EC50/90 values fit to published data? Provide references.
102 and 175 Please indicate the time between infection and adding the compound
124 also explain how the HSA model works and what it shows.
176 Please also include the raw data, possibly similar to Figure 1. Can you calculate a “new EC50/90”?
Fig 2 use consistent labels PF-07321332 or nirmatrelvir. What does the white square indicate? Why does 5.7 µM rem and 0.5 µM nirm lead to higher synergy than with 0.9 µM nirm? why are concentrations in A and B not the same?
189 give more examples for log titer reductions: also for the compounds alone
Fig 3: does the data correlate with Fig 2? How does 3.1 + 4.5 compare to 0.8 + 2.2, as the HSS index seems to be higher in the latter? Why was only data of 72 h shown?
249 please state that (if) it was unclear if negativity was reached in between
350 another study found enhanced activity: DOI: 10.1126/sciadv.add7197.
365 could synergy not simply be reached due to targeting two instead of only one viral enzymes?
389 unfortunately, there is no proof that negativity wouldn’t have been reached without treatment
Author Response
Gidari et al provide in vitro data showing the synergistic effects of remdesivir and nirmatrelvir and provide a case report where this combination might have contributed to the viral clearance in an immunosuppressed covid-19 patient. The amount of data provided is low, but appears valid. However, the methodology needs to be explained and referenced more thoroughly. The interpretation that this combination was required for negativization could be weakened. Nonetheless, since this combination is clinically applied in patients, the in vitro analysis provided here gives important background information.
Specific points
62 Current therapies also include monoclonal antibodies. Also the patient in the case report received several antibody treatments. This should be added here.
Thank you for the suggestion, the sentence has been corrected.
66-70 provide references
The reference has been added.
88-90 Provide GISAID IDs
GISAID IDs are not available because genome sequenced has been done in another centre.
106 in the provided reference [13] authors use Vero-GFP cells to monitor CPE. When working with other cells they use MTS instead of MTT. Vero cells do not express TMPRSS2 so the CPE is low, making determination via viability assays not very effective. However, there is differences between labs and CPE is very prominent in the author’s hands. To confirm this, microscopy images of CPE should be provided.
Thank you for the suggestion. In their experiments, Abdelnabi et al. used a GFP-expressing Vero E6 cell line that does not express TMPRSS2 like our cells. This cell line uses a different mechanism of viral entry than TMPRSS2-expressing cells, so the model is suitable for testing compounds that have different targets than cellular entry. Furthermore, we used MTT to evaluate the metabolic activity of Vero cells which is equivalent to MTS, since the only difference between the two methods is the addition of solubilizing solution for formazan crystals. For these reasons, the effect of antivirals was tested following reference [17] with some modifications. The sentence has been modified (lines 109)
The microscopy images of CPE has been added as Supplementary Figure 2.
115 Was the inoculum also removed (wash control)? Or do the 48 h samples still contain the initially added virus?
The sentence “Cells were incubated with 0.1 MOI of virus for 1 h, then the supernatant was removed, and compounds added” has been added in lines 132-133
Fig 1 Please also normalize this Figure and show as %infection or as %viability recovery, so the EC50/90 can be easily estimated from the graph. Also include the values obtained in absence of virus. Consider to show the x-axis as log scale. Do the determined EC50/90 values fit to published data? Provide references.
We added dashed lines on the Figure to underline EC50/90 (See Supplementary Figure 1). It is not possible to show the x-axis as a log scale because it would not be possible to plot the 0 (not treated cells). To the best of our knowledge, there are no data of remdesivir EC50 and EC90 with MTT assay, so it is not possible to compare EC50 and EC90 obtained with other methods. However, such as our results, published data showed micromolar values of EC50 and EC90 (DOI: 10.1016/j.antiviral.2020.104878).
102 and 175 Please indicate the time between infection and adding the compound
Thank you for the suggestion, the time between infection and adding the compound has been added (lines 132-133)
124 also explain how the HSA model works and what it shows.
Thank you for the suggestion, the HSA model was explained in lines 141-145.
176 Please also include the raw data, possibly similar to Figure 1. Can you calculate a “new EC50/90”?
We added the raw data panel (See new Figure 1). It is not possible to calculate EC50/90 for the combination because each concentration of one of the single compounds is combined with different concentrations of the other compound. The best way to calculate the interaction between the drugs is to calculate a synergy score.
Fig 2 use consistent labels PF-07321332 or nirmatrelvir. What does the white square indicate? Why does 5.7 µM rem and 0.5 µM nirm lead to higher synergy than with 0.9 µM nirm? why are concentrations in A and B not the same?
PF-07321332 has been substituted with nirmatrelvir. The white square indicates the most synergistic area, for simplicity, it has been removed. Remdesivir 5.7 µM combined with nirmatrelvir 0.5 or 0.9 µM highly improved its activity, the difference between the two concentrations of nirmatrelvir is not significant. Concentrations in A and B are not the same because EC50 and EC 90 are not the same at 48 and 72 h (see supplementary Figure 1, previously named Figure 1).
189 give more examples for log titer reductions: also for the compounds alone
Thank you for the suggestion. The examples have been added in lines 228-233.
Fig 3: does the data correlate with Fig 2? How does 3.1 + 4.5 compare to 0.8 + 2.2, as the HSS index seems to be higher in the latter? Why was only data of 72 h shown?
We selected 2 concentrations of the combination from the most synergistic area and 1 higher to find if the combination could reach eradication (< 50 PFU/ml is the limit of the method), confirming that it was reached (this aspect is now specified in line… and in figure 2 caption). We selected 72 h as the time of incubation because the results of viability tests are more homogeneous as you can see in Figure 1 (previously named Figure 2).
249 please state that (if) it was unclear if negativity was reached in between
Negativity was never reached before 2nd November 2022, see line 325 and the dashed line of Figure 3 (previously named Figure 5).
350 another study found enhanced activity: DOI: 10.1126/sciadv.add7197.
Thank you for the report. The paper has been discussed (see lines 415-419).
365 could synergy not simply be reached due to targeting two instead of only one viral enzymes?
Yes, it is possible, but in that case, it is more probable to see an additive effect, a synergistic effect implies that the compounds interact in some way.
389 unfortunately, there is no proof that negativity wouldn’t have been reached without treatment
Of course, it is not possible to prove it, but she remained positive for 9 months without treatment, at the same time nothing could warrantee that this severely immunocompromised patient could eradicate the SARS-CoV-2 infection without medical treatment. Of note, the combination remdesivir/nirmatrelvir may have played a role in the interstitial pneumonia resolution and may have reduced the viral load, but maybe it was too short to eradicate the infection. Indeed, the patient eradicated the infection later and only under prolonged therapy with nirmatrelvir/ritonavir.
Reviewer 4 Report
The authors have attempted to study the impact of remdesivir-nirmatrelvir combination therapy against SARS-CoV-2 in vitro and have documented a case study which demonstrates its potential utility in treatment of COVID.
I feel this manuscript should be revised, and the case study removed to focus on the in vitro aspect of the work. Given number and variety of therapies administered to patient, the link to the remdesivir-nirmatrelvir combination and favourable patient outcome cannot be made. Presenting the case study alongside the in vitro data feels somewhat misleading.
I offer the following comments for consideration:
Title
1. I feel the title could provide more detail without overly extending the word count. Please consider revision, particularly if the case study is removed.
Introduction
1. Nicely refenced – puts the research in to the context of the field in an unbiased fashion.
2. Page 2, Line 64
a. Please expand this sentence describe the reason for withdrawal of molnupiravir:
i. ‘However, from March 64 2023 molnupiravir is no longer available for prescription, as “Agenzia Italiana del Far- 65 maco” (AIFA) stated[9]’
3. Page 2, Line 66
a. This sentence is difficult to follow, please revise:
i. ‘Similar drugs were initially used against HIV and for this reason, 66 it was possible to develop new ones in a very short time thanks to the knowledge already 67 acquired and the means available for scientific research. ‘
Materials and Methods
1. Methods are appropriate and referenced, where necessary.
Results
1. Figure 1
a. Replot figures with x-axis in log-scale
b. For clarity and transparency, please show the 3 biological replicates on plot and include error bars showing the SD for each of the three technical replicates
2. Figure 2
a. Please show the SyngeryFinder plot which includes numerical values across the matricies
b. 3D plot may also aid visualisation
3. Figure 3 and 4
a. Please combine these figures.
b. Please perform appropriate SynergyFinder analyses for each variant
4. The case study is well-described. However, due to the wide-array of treatments administered, I do not feel it can be described as a case-study of remdesivir-nirmatrelvir combination therapy. The outcome cannot be attributed to this one therapeutic strategy.
Discussion
1. Page 9, Line 339
a. Not referenced. Other more-relevant combination therapies have demonstrated activity in vitro – consider citing other sources dealing with Vero-based models assessing remdesivir-based combination therapies.
2. Page 10, Line 361
a. The data presented do not demonstrate true synergy against all variants tested. To make this claim for all variants please perform the appropriate SynergyFinder analyses.
3. Limitations require further expansion – specifically, the authors should explain why the model is likely to ‘overestimate the antiviral efficacy’ and explain what the associated limitations of the ‘non-human cell lines’ are and how this may impact the observations made.
Author Response
The authors have attempted to study the impact of remdesivir-nirmatrelvir combination therapy against SARS-CoV-2 in vitro and have documented a case study which demonstrates its potential utility in treatment of COVID.
I feel this manuscript should be revised, and the case study removed to focus on the in vitro aspect of the work. Given number and variety of therapies administered to patient, the link to the remdesivir-nirmatrelvir combination and favourable patient outcome cannot be made. Presenting the case study alongside the in vitro data feels somewhat misleading.
We believe that the case report is important because makes it possible to correlate the in vitro results with clinical practice for different reasons. Firstly, one of the SARS-CoV-2 strains tested (BA.1 variant) was isolated from the patient of the described case report. Furthermore, it is not possible to establish if the combination treatment is related to a good outcome, but also in other reports, patients underwent multiple treatments. This does not exclude the importance of antiviral treatment in clinical cases where viral disease is the most critical problem. It also helps to confirm the safety of this treatment.
I offer the following comments for consideration:
Title
- I feel the title could provide more detail without overly extending the word count. Please consider revision, particularly if the case study is removed.
Thank you for the suggestion, the title has been modified.
Introduction
- Nicely refenced – puts the research in to the context of the field in an unbiased fashion.
Thank you.
- Page 2, Line 64
- Please expand this sentence describe the reason for withdrawal of molnupiravir:
- ‘However, from March 64 2023 molnupiravir is no longer available for prescription, as “Agenzia Italiana del Far- 65 maco” (AIFA) stated[9]’
Thank you for the suggestion. The sentence has been expanded (see lines 70-71)
- Page 2, Line 66
- This sentence is difficult to follow, please revise:
- ‘Similar drugs were initially used against HIV and for this reason, 66 it was possible to develop new ones in a very short time thanks to the knowledge already 67 acquired and the means available for scientific research. ‘
Thank you for the suggestion. The sentence has been simplified.
Materials and Methods
- Methods are appropriate and referenced, where necessary.
Thank you for the comment.
Results
- Figure 1
- Replot figures with x-axis in log-scale
It is not possible to show the x-axis as a log scale because it would not be possible to plot the 0 (not treated cells).
- For clarity and transparency, please show the 3 biological replicates on plot and include error bars showing the SD for each of the three technical replicates
Thank you for the suggestion, the figure has been modified (see Supplementary Figure 1).
- Figure 2
- Please show the SyngeryFinder plot which includes numerical values across the matricies
- 3D plot may also aid visualisation
Thank you for the suggestions, the figure has been modified.
- Figure 3 and 4
- Please combine these figures.
Thank you for the suggestion, the figures have been combined (see Figure 2 A, B, C).
- Please perform appropriate SynergyFinder analyses for each variant
Round 2
Reviewer 2 Report
Authors have made sufficient changes and revised manuscript is improved.
Reviewer 4 Report
Much improved - happy with the manuscript in its current form.